# Nrf2-Related Therapeutic Effects of Curcumin in Different Disorders

**DOI:** 10.3390/biom12010082

**Published:** 2022-01-05

**Authors:** Soudeh Ghafouri-Fard, Hamed Shoorei, Zahra Bahroudi, Bashdar Mahmud Hussen, Seyedeh Fahimeh Talebi, Mohammad Taheri, Seyed Abdulmajid Ayatollahi

**Affiliations:** 1Department of Medical Genetics, School of Medicine, Shahid Beheshti University of Medical Sciences, Tehran 16666-63111, Iran; s.ghafourifard@sbmu.ac.ir; 2Department of Anatomical Sciences, Faculty of Medicine, Birjand University of Medical Sciences, Birjand 9717853577, Iran; h.shoorei@gmail.com; 3Department of Anatomical Sciences, Faculty of Medicine, Tabriz University of Medical Sciences, Tabriz 5166-15731, Iran; z.bahroudi@gmail.com; 4Department of Pharmacognosy, College of Pharmacy, Hawler Medical University, Erbil 44001, Kurdistan Region, Iraq; Bashdar.Hussen@hmu.edu.krd; 5Department of Pharmacology, College of Pharmacy, Birjand University of Medical Sciences, Birjand 9717853577, Iran; s.fahimehtalebi@gmail.com; 6Institute of Human Genetics, Jena University Hospital, 07743 Jena, Germany; 7Phytochemistry Research Center, Shahid Beheshti University of Medical Sciences, Tehran 16666-63111, Iran

**Keywords:** Nrf2, curcumin, disorders, cancer

## Abstract

Curcumin is a natural polyphenol with antioxidant, antibacterial, anti-cancer, and anti-inflammation effects. This substance has been shown to affect the activity of Nrf2 signaling, a pathway that is activated in response to stress and decreases levels of reactive oxygen species and electrophilic substances. Nrf2-related effects of curcumin have been investigated in different contexts, including gastrointestinal disorders, ischemia-reperfusion injury, diabetes mellitus, nervous system diseases, renal diseases, pulmonary diseases, cardiovascular diseases as well as cancers. In the current review, we discuss the Nrf2-mediated therapeutic effects of curcumin in these conditions. The data reviewed in the current manuscript indicates curcumin as a potential activator of Nrf2 and a therapeutic substance for the protection of cells in several pathological conditions.

## 1. Introduction

Alternatively named as Nrf2, nuclear factor erythroid-derived 2-like 2 (NFE2L2) is a transcription factor that induces expression of genes in response to stress and decreases levels of reactive oxygen species (ROS) and electrophilic substances, thus being regarded as a modality for the prevention of chronic disorders [1,2].

The natural polyphenol curcumin (Figure 1) has been found to exert many of its beneficial effects through activation of the Nrf2 pathway [3]. Being extracted from the rhizome of Curcuma longa, curcumin has antioxidant, antibacterial, anti-cancer, and anti-inflammation effects.

The poor bioavailability of curcumin has a major obstacle in its clinical application. Several investigations have shown very low or untraceable concentrations of curcumin in the circulation and extraintestinal tissues. This finding is due to poor absorption of curcumin, its fast metabolism, instability of curcumin, and fast systemic removal [4]. Animal studies have shown excretion of more than 90% of oral curcumin in the feces [5]. The main applied strategies to enhance the bioavailability of curcumin are the application of adjuvants such as piperine, making liposomal curcumin formulations, curcumin nanoparticles, curcumin phospholipid complex as well as structural analogs [6].

The microbiota of the intestine can change curcumin in different metabolism pathways, such as making active metabolites which can induce local and systemic impacts. Moreover, they can reduce the heptadienone backbone and dimethylate this substance [7,8].

This natural substance has potential applications in a variety of disorders, such as neurodegenerative diseases, renal diseases, and diabetes mellitus. In addition, its effects on the Nrf2 signaling pathway potentiate it as a protective factor against oxidative damage [3]. Curcumin-mediated induction of Nrf2 signaling relies on four putative mechanisms, namely suppression of Keap1, regulation of activity of upstream mediators of Nrf2, regulation of expression of Nrf2 and its targets, and enhancement of Nrf2 nuclear translocation [3]. Nrf2-related effects of curcumin have been investigated in different contexts, including gastrointestinal disorders, diabetes mellitus, nervous system diseases, renal diseases, pulmonary diseases, cardiovascular diseases as well as cancers. In the current review, we discuss the Nrf2-mediated therapeutic effects of curcumin in these conditions.

## 2. Digestive System Diseases

Tetrahydrocurcumin (THC) and octahydrocurcumin (OHC) are the primary and final hydrogenated metabolites of curcumin. These curcumin metabolites have high antioxidant activities and can ameliorate acetaminophen-induced hepatotoxicity and amend histopathological changes. In addition, these curcumin metabolites could restore antioxidant conditions in the liver by decreasing MDA and ROS levels and increasing GSH, SOD, CAT, and T-AOC levels. Moreover, they have decreased activity and expression of CYP2E1, induced the Keap1-Nrf2 pathway, and promoted expressions of Nrf2-targeted genes. The latter effect has been exerted through suppression of Keap1 expression and inhibition of Keap1/Nrf2 interaction [10]. Another study has shown the effect of curcumin on decreasing amounts of hepatic steatosis and inflammatory responses in adult Sprague–Dawley rats. Moreover, curcumin could decrease serum levels of ALT, AST, and lipids and amend insulin resistance. Serum and hepatic amounts of TNF-α, IL-6, and MDA have also been decreased following the administration of curcumin. Curcumin could also increase Nrf2 levels in nuclei of liver cells, indicating that this substance has a potential effect in the prevention and amelioration of nonalcoholic steatohepatitis through a reduction in lipid levels and inflammatory responses, improvement of insulin resistance, and enhancement of antioxidants, possibly through induction of Nrf2 [11].

Curcumin has also been shown to prevent hepatic injury induced by post-intrahepatic inoculation of trophozoites. Macroscopic and microscopic evaluations in animal models have shown that this substance can decrease ALT, ALP, and γ-GTP activities. Curcumin could also ameliorate the amoebic damage-induced reduction in glycogen content and suppress NF-κB activity and IL-1β levels while inducing a tendency toward up-regulation of Nrf2 production [12]. Moreover, curcumin could attenuate ethanol-induced hepatic steatosis via modulation of the Nrf2/FXR axis [13]. Table 1 summarizes Nrf2-related therapeutic effects of curcumin in gastrointestinal diseases.

## 3. Ischemia-Reperfusion (IR) Injury

An antioxidant mono-carbonyl analog of curcumin (MACs) has been shown to exert a protective effect against ischemia/reperfusion (I/R)-induced cardiac injury. This structurally modified formulation of curcumin does not have the β-diketone moiety and possesses improved in vitro stability and better in vivo pharmacokinetic profiles. Based on the results obtained from the *in vitro* cell-based screening experiments, pre-treatment of H9c2 cells with a certain curcumin analog could activate the Nrf2 signaling pathway, reduce H_2_O_2_-induced up-regulation of MDA and SOD levels, suppress TBHP-induced cell death, and reduce the activity of Bax/Bcl-2–caspase-3 axis. Subsequent studies in animal models of myocardial I/R have also verified the effect of curcumin on the reduction in infarct size and myocardial apoptosis [27]. Another study has shown that the combination of the immunomodulatory drug dimethyl fumarate (DMF) and curcumin has a prominent ameliorative effect in I/R-induced hepatic injury, as is evident by a remarkable decrease in serum ALT and AST activity and improvement of histopathological features. The protective effects of DMF have been shown to be exerted through activation of Nrf2/HO-1 signaling and enhancement of GSH and TAC levels. Curcumin could influence levels of inflammatory markers and infiltration of neutrophils. Besides, curcumin has enhanced DMF-induced Nrf2/HO-1 activation [28]. Finally, another experiment has shown that curcumin could reduce neurological dysfunction and brain edema after cerebral I/R through elevation of Nrf2 and down-regulation of NF-κB [29] (Figure 2). Table 2 shows Nrf2-related therapeutic effects of curcumin in IR injury.

## 4. Diabetes Mellitus and Its Related Complications

Curcumin has a synthetic derivative, namely (2E,6E)-2,6-bis(2-(trifluoromethyl)benzylidene) cyclohexanone or C66. This curcumin derivative has been shown to ameliorate diabetes-induced pathogenic alterations in the aorta through activation of Nrf2. This substance could amen to diabetes-associated oxidative stress in the aorta and reverse the effects of diabetes on apoptosis, inflammation, and fibrosis of the aorta. While either C66 or JNK2 deletion could activate Nrf2, C66 had no additional influence on diabetic aortic injury or Nrf2 activity without JNK2. Taken together, C66 can protect against diabetes-associated pathological alterations in the aorta through suppression of JNK2 and enhancement of Nrf2 levels and function [30]. Another study has shown that curcumin analog A13 can reduce the histological changes in the myocardial tissues of diabetic rats. A13 is a hydrosoluble mono-carbonyl analogue of curcumin with the following formulation: (2E,5E)-2,5-bis(4-(3-(dimethylamino)-propoxy)benzylidene)cyclopentanone. Curcumin and A13 could also decrease MDA levels and enhance SOD activity in this tissue through activation of the Nrf2/ARE pathway [31]. The protective effect of curcumin against diabetes-related retinopathy is also related to its role in the activation of the Nrf2/HO-1 axis [32]. Table 3 summarizes Nrf2-related therapeutic effects of curcumin in diabetes.

## 5. Nervous System Diseases

Curcumin has been shown to ameliorate traumatic brain injury (TBI)-induced abnormal changes in the brain, as evident by amendment of the water content of the brain, oxidative stress, neurological severity score, and apoptosis of neurons. The anti-apoptotic effects of curcumin have been verified through the observed elevation of Bcl-2 levels and reduction in cleaved caspase-3 levels. Notably, curcumin could increase the nuclear translocation of Nrf2, enhance levels of HO1 and NADPH: NQO1, and preclude the reduction in antioxidant enzymes activity. Taken together, curcumin has been found to enhance the activity of antioxidant enzymes and decrease brain injury, most probably through enhancing the activity of the Nrf2/ARE axis [36]. Another study has shown that the administration of curcumin can reduce ipsilateral cortex injury, infiltration of neutrophils, and activation of microglia in animal models of TBI. These effects have led to improvement of neuron survival and reduction in TBI-associated apoptosis and degeneration. Nrf2 has been found to be the main mediator of these effects since Nrf2 deletion has diminished the neuroprotective impact of curcumin [37]. Curcumin has also been shown to ameliorate radiation-induced cerebral injury through up-regulation of Nrf2 [38]. In addition, curcumin could amend motor dysfunction and enhance the activity of tyrosine hydroxylase in the substantia nigra pars compacta of rotenone-injured rats. This natural substance could increase GSH levels and decrease ROS activity and MDA level. The effects of curcumin in amelioration of rotenone-associated oxidative damage in dopaminergic neurons have been shown to be exerted through activation of the Akt/Nrf2 axis [39]. Table 4 shows Nrf2-related therapeutic effects of curcumin in nervous system disorders.

## 6. Renal Diseases

Co-administration of thymoquinone and curcumin has been shown to ameliorate cisplatin-induced kidney toxicity. These two substances have synergistic effects in the reduction of cisplatin-induced apoptosis in HEK-293 cells. They could also amend the effects of cisplatin on antioxidant enzyme concentrations and mitochondrial ATPases. Akt, Nrf2, and HO-1 levels have been increased by thymoquinone and curcumin. Moreover, these agents could decrease cleaved caspase 3 and NF-κB levels in kidney homogenates [46]. An animal study has shown that the impact of curcumin in the treatment of chronic kidney disease (CKD) is mediated via the Keap1/Nrf2 axis [47]. However, a randomized, double-blind placebo-controlled clinical trial in patients with nondiabetic or diabetic proteinuric CKD has shown no significant effect of curcumin on proteinuria, estimated glomerular filtration rate, or lipid profile. Yet, curcumin could attenuate lipid peroxidation in plasma of patients with nondiabetic proteinuric CKD and improve antioxidant activity in patients with diabetic proteinuric CKD [48]. Table 5 summarizes the Nrf2-related therapeutic effects of curcumin in renal disorders.

## 7. Pulmonary Diseases

An experiment in lung mesenchymal stem cells (LMSCs) has shown that curcumin could decrease ROS levels while increasing mitochondrial membrane potential levels. Moreover, curcumin has decreased expression levels of cleaved caspase-3, enhanced Nrf2 and HO-1 levels, and increased Bcl-2/Bax and p-Akt/Akt ratios. Taken together, curcumin has been shown to protect against H_2_O_2_-associated LMSC damage via regulation of the Akt/Nrf2/HO-1 signaling [50]. Another study has shown that the potential effect of curcumin in the prevention of high altitude pulmonary edema is exerted via up-regulation of Nrf2 and HIF1-α [51]. Moreover, it has been shown to attenuate airway inflammation in asthma via activating Nrf2/HO-1 axis [52]. Table 6 shows Nrf2-related therapeutic effects of curcumin in pulmonary disorders.

## 8. Other Conditions

Therapeutic effects of curcumin have also been investigated in a variety of disorders such as osteoporosis, temporomandibular joint osteoarthritis, muscle damage, skin damage, heat- or H_2_O_2_-induced oxidative stress, and cystopathy (Table 7). Moreover, curcumin could effectively inhibit quinocetone-induced apoptosis via inhibiting the NF-κB and activating Nrf2/HO-1 axis [54]. Besides, curcumin could protect osteoblasts against oxidative stress-induced dysfunction through the GSK3b-Nrf2 axis [55]. Table 8 summarizes Nrf2-related therapeutic effects of curcumin in diverse conditions.
biomolecules-12-00082-t007_Table 7Table 7Nrf2-related therapeutic effects of curcumin in other conditions.Type of DiseaseAnimals Dose RangeCell LineTargets andOther PathwaysFunctionRef.---L02HO-1, NF-κB, iNOS, HO-1, Caspase-3/9CUR via inhibiting the NF-κB and activating Nrf2/HO-1 axis could effectively inhibit quinocetone (QCT) induced apoptosis.[54]Osteoporosis--MC3T3-E1ALP, OCN, COLI, Runx2CUR through the GSK3β-Nrf2 axis could protect osteoblasts against oxidative stress-induced dysfunction.[55]---hPDLSCsAKT, PI3K, ALP, COL1, RUNX2CUR via the PI3K/AKT/Nrf2axis could promote osteogenic differentiation of hPDLSCs.[56]Temporomandibular Joint Osteoarthritis (TMJ OA)--ChondrocytesARE, HO-1, SOD2, NQO-1, IL-6, iNOS, MMP-1/3/pCUR via the Nrf2/ARE axis could inhibit oxidative stress, inflammation, and the matrix degradation of TMJ inflammatory chondrocytes.[57]Muscle DamageMale Wistar rat100 mg/kg, orally, daily, 6 weeks-NF-κB, GLUT4, HO-1, PGC-1α, SIRT1, TRX-1CUR via regulating the NF-κB and Nrf2 pathways could prevent muscle damage.[58]Skin DamageFemale ICR mice0.1–1 μmol, topicallyJB6, 293T, MEFsHO-1, Cullian3, Rbx1CUR via the Keap1 cysteine modification could induce stabilization of Nrf2.[59]Heat-Induced Oxidative Stress--CEFARE, SOD1, MAPK, ERK, JNK, p38CUR via activating the MAPK-Nrf2/ARE axis could inhibit heat-induced oxidative stress in chicken fibroblasts cells.[60]---Mouse cortical neuronal cells, 293T, MEFsHO-1, NQO1, GST-mu1,p62, NDP52, CUR via the PKCδ-mediated p62 phosphorylation at Ser351 could activate the Nrf2 pathway.[61]H_2_O_2_-Induced Oxidative Stress--HTR8/SVneoHO-1, GCLC, GCLM, NQO1, SLC2A1/3, Bax, Bcl-2, Caspase-3CUR via activating the Nrf2 could protect HTR8/SVneo cells from H_2_O_2_-induced oxidative stress.[62]---SKBR3,U373HO-1, p62, SQSTM1In response to Zn(II)–curcumin complex, p62/SQSTM1/Keap1/Nrf2 axis could reduce cancer cells death-sensitivity.[63]Zearalenone (ZEA)-Induced Apoptosis And Oxidative Stress--TM3PTEN, HO-1, Bip, AKT, Bax, Bcl-2, JNK, Caspase-3/9/12CUR by modulating the PTEN/Nrf2/Bipaxis could inhibit ZEA-induced apoptosis and oxidative stress.[64]---HepG2-C8HO-1, UGT1ACombining low doses of CUR and sulforaphane via Nrf2 could play a role in the prevention of several types of cancer.[65]Cisplatin-Induced Drug Resistance--A549/CDDPSQSTM1(P62), LC3-I, LC3-II, NQO1CUR via the Keap1/p62-Nrf2axis could attenuate CDDP-induced drug-resistance in A549/CDDP cell.[66]Cisplatin-Induced Bladder CystopathyFemale SD rats6 mg/kg,5 consecutive daysRBSMCs, SV-HUC-1, ATCC, Manassas, VANGF, HO-1CUR via targeting NRF2 could ameliorate cisplatin-induced cystopathy.[67]Pain Male Swiss mice3, 10, 30 mg/kg, subcutaneously, 1 h before stimulation-NF-ĸB, HO-1, TNF-α, IL-10, IL-1βCUR via reducing NF-κB activation and increasing Nrf2 expression could inhibit superoxide anion-induced inflammatory pain-like behaviors.[68]EndotoxemiaMale Wistar rats25,50, and 100 mg/kg, orally, 2 consecutive days-TNF-α, IL1-βCUR via modulating the activity of Nrf2 could prevent LPS-induced sickness behavior and fever possibly.[69]Cadmium-Induced Testicular InjuryKunming mice50 mg/kg, I.P., 10 days-GSH-Px, γ-GCSCUR by activating the Nrf2/ARE axis could protect against cadmium-induced testicular injury.[70]Oxidative Damage--RAW264.7HO-1, GCLC, GLCMCUR via activating the Nrf2-Keap1 pathway and increasing the activity of antioxidant enzymes could attenuate oxidative stress in RAW264.7 cells.[71]Nasal Diseases--Nasal fibroblastsHO-1, ERK, SOD2CUR via activating of the Nrf2/HO-1 axis could reduce ROS production caused by urban particulate matter (UPM) in human nasal fibroblasts.[72]Aβ25-35-Induced Oxidative Damage--PC12HO-1, Bcl-2, Bax, Cyt-cCUR analogs via the Keap1/Nrf2/HO-1 axis could attenuate Aβ25-35-induced oxidative stress in PC12 cells.[73]Thyroid dysfunctionMale Wistar rats30 mg/kg, orally, daily, 30 days-NF-ĸB, AKT, mTOR, SOD1, SOD2CUR/vitamin E via modulating the Nrf2 and Keap1 function could reduce oxidative stress in the heart of rats.[74]
Figure 2Oxidative stress formed by a number of disorders or external factors such as chemical drugs, heat stress, and so on could induce both extrinsic and intrinsic apoptotic pathways [75]. However, during abnormal conditions, endoplasmic reticulum (ER) stress could also activate the intrinsic apoptotic pathway [76], leading to cell death. On the other hand, disruption of Bax/Bcl-2 balance by mitochondrial dysfunction leads to ROS elevation [75]. Afterward, ROS activate the NF-κB signaling pathway, subsequently increasing the release of inflammatory cytokines. Interestingly, antioxidant sources-such as curcumin-could decrease ROS production and cell death rate, finally. In this regard, curcumin via activating the Nrf2 pathway could increase the levels of cellular antioxidants [43,48], such as SOD, GPx, and CAT, and by activating the expression of HO-1 (Heme Oxygenase-1), as an Nrf2-regulated gene which is involved in the prevention of vascular inflammation, could directly or indirectly decrease the generation of ROS as well as inflammation [28,54]. On the one hand, it has been reported that curcumin via activating the ERK and MAPK could ease the oxidative damage [32,60].
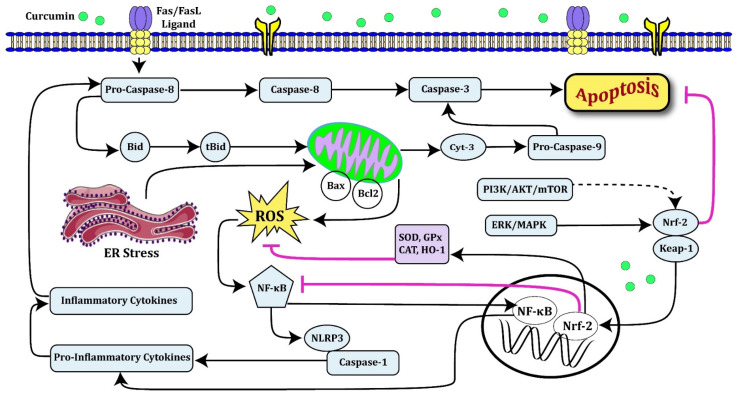


## 9. Cancers

The beneficial effects of curcumin have also been investigated in malignant conditions. In animal models of ovarian cancer, curcumin could prevent epithelial-mesenchymal transition (EMT)-mediated progression through modulation of Nrf2/ETBR/ET-1 axis [77]. In colorectal cancer cells, curcumin could affect multidrug resistance via modulation of Nrf2 [78]. Curcumin can inhibit the proliferation of breast cancer cells via Nrf2-mediated down-regulation of Fen1 [79].

A single study in an immortalized lymphoblastoid cell line has shown that treatment of cells with a proapoptotic dose of curcumin can lead to nuclear accumulation of Nrf2 and the expression of Nrf2 targets at early phases, while at late phases it total and nuclear protein levels of Nrf2 have been decreased and Nrf2 targets have been down-regulated in the absence of p53 activity. Thus, apoptosis-associated inactivation of Nrf2 can happen in a p53 dysfunctional context [80]. Table 8 shows Nrf2-related therapeutic effects of curcumin in cancers.
biomolecules-12-00082-t008_Table 8Table 8Nrf2-related therapeutic effects of curcumin in cancers.Type of DiseaseType of CurcuminAnimals Dose RangeCell LineTargets andOther PathwaysFunctionRef.Ovarian Carcinoma (OC)CURFemale Wistarrat100 mg/kg, orally, daily, 4 weeksSKOV3ETBR, ET-1, Caspase-3/9, Bax, Bcl-2, N-cadherin, E-cadherin, VimentinCUR via the Nrf2/ETBR/ET-1 axis could prevent EMT-mediated OC progression.[77]LymphoblastomaCUR--CL-45p53, Caspase-3/9, PARP, HMOX1CUR during oxidative stress-induced apoptosis could induce p53-independent inactivation of Nrf2.[80]Prostate Cancer (PCa)F10,E10--TRAMP-C1,HepG2-C8ARE, HO-1, UGT1A1, NQO1, HDAC7,H3, DNMT3a, DNMT3bCurcumin derivatives could reactivate Nrf2 in TRAMP C1 cells.[81]Colorectal Cancer(CRC)CUR--HCT-8/5-Fu, HCT-8NQO1, Bcl-2, BaxCUR via the Nrf2 could affect multidrug resistance (MDR) in human CRC.[78]Breast Cancer(BC)CUR--MCF-7Fen1, AKR1B10, AKR1C1/3CUR via Nrf2-mediated down-regulation of Fen1 could inhibit the proliferation of BC cells.[79]

## 10. Discussion

Curcumin is a natural substance that has been shown to increase nuclear levels of Nrf2 and enhance the biological function of this nuclear factor through interacting with Cys151 in Keap1 [82]. This substance is an important therapeutic modality for a variety of oxidative stress-related disorders such as diabetes mellitus, brain disorders, cardiovascular disorders, and malignancies. In addition to modulating antioxidant enzymes and inflammatory responses, curcumin can affect the activity of NF-κB. Other pathways modulated by curcumin should also be investigated in different contexts. This information would help in better understanding the mechanism of the therapeutic effects of curcumin.

The impact of curcumin on Nrf2 expression has been vastly investigated in cell lines as well as animal models. In cell lines, both curcumin and its analogs could exert functional effects through modulation of Nrf2 expression.

Experiments in animal models of different disorders have shown its beneficial effects on animal health and amelioration of pathological events during the course of malignant or non-malignant disorders.

A single randomized, double-blind placebo-controlled clinical trial in patients with CKD has shown no significant impact of curcumin on Nrf2 expression, despite its effectiveness in the improvement of antioxidant activity [48]. Thus, it is necessary to conduct sufficient studies in human subjects to verify the results of in vitro and animal studies.

Taken together, Nrf2-related effects of curcumin have not been thoroughly and systematically investigated in human subjects. Thus, there is no predictor for assessment of response to this natural substance in different diseases. Moreover, the Nrf2-related effects of this substance have been less investigated in neoplastic conditions. Based on the complexity of gene networks in these conditions, it is necessary to find exact targets of curcumin in cancers.

## Figures and Tables

**Figure 1 biomolecules-12-00082-f001:**
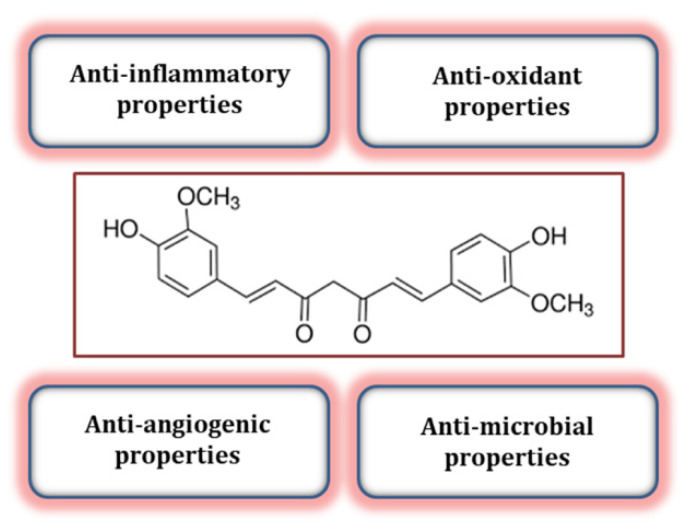
Chemical structure of curcumin (1,7-bis(4-hydroxy-3-methoxyphenyl)-1,6-heptadiene-3,5-dione). Curcumin which is found in the rhizome of Curcuma longa (turmeric) as a strong natural polyphenol, has a number of biological activities, including antioxidant (could inhibit ROS-generating enzymes) and anti-inflammatory (could block NF-κB activation) properties, anti-cancer, anti-obesity, and anti-infertility effects [9].

**Table 1 biomolecules-12-00082-t001:** Nrf2-related therapeutic effects of curcumin in gastrointestinal diseases (I.P.: intraperitoneal).

Type of Disease	Type of Curcumin	Animals	Dose Range	Cell Line	Dose Range	Targets and Other Pathways	Function	Ref.
Liver Injury	OHC, THC, CUR	Male Kunming mice	OHCandTHC: 25, 50, and 100 mg/kg, CUR: 100 mg/kg, I.P, pretreatment for 30 min	-	-	CYP2E1, GCLC, GCLM, NQO1, HO-1	OHC and THC, via restoring antioxidant status, inhibiting CYP2E1, and activating the Keap1-Nrf2 pathway could protect the liver against APAP toxicity.	[10]
Liver Injury	CUR	SD rat	50 mg/kg, orally, daily, 6 weeks	-	-	HO-1, TNF-α, IL-6,	CUR via the Nrf2 pathway could attenuate oxidative stress and liver inflammation in rats with NASH.	[11]
Acute Liver Injury (ALI)	CUR	Adult C57BL/6 mice	50, 100, and 200 mg/kg, orally, daily, 7 days	-	-	HO-1, Caspase-3/9, TNF-α, IL-1β, TGF-β1/Smad3	CUR by activating Nrf2/HO-1 and inhibiting TGF-β1/Smad3 could protect against CCl4-induced ALI.	[14]
Liver Injury	CUR	Male Kunming mice	50 mg/kg, Orally, 2 h before HgCl_2_ injection	L02 hepatocytes	5 μM	HO-1, Nqo1, Il-1β, TNF-α, Caspase-1	CUR by Nrf2/HO-1 pathway and modulate cytochrome P450 could improve mercuric chloride-induced liver injury.	[15]
Acute Liver Injury (ALI)	CUR	Male SD rat	5 mL/kg, orally, daily, starting 3 days before LPS/D-GalN treatment	-	-	TNF-α, NF-κB, HO-1, NQO-1, AKT, p65	CUR via inhibiting NF-κB and activating Nrf2 could attenuate lipopolysaccharide/D-galactosamine-induced ALI.	[16]
Oxaliplatin (OXA)-Induced Liver Injury	CUR	BALB/CJ mice	100 mg/kg, daily, orally, 8 weeks	-	-	HO-1, NOQ1, CXCL1,CXCL2, MCP-1, PAI-1,	CUR via activating the Nrf2 could attenuate OXA-induced liver injury.	[17]
Amoebic Liver Abscess (ALA)	CUR	Hamster	150 mg/kg, orally, daily during 10 days before infection	-	-	HO-1, NF-κB, IL-1β	CUR via Nrf2/HO-1 axis could play a role in providing hepato-protection against ALA.	[12]
Alcoholic Liver Disease (ALD)	CUR	Male SD rats	100, 200, and 400 mg/kg, orally, 9 weeks	LO2	10–40 µM	FXR, TNF-α, NF-κB, PPAR-α	CUR via modulating Nrf2/FXR axis could attenuate ethanol-induced hepatic steatosis.	[13]
ALD	CUR	male Wistar rats	50 mg/kg, daily, orally, Third to the fourth week of the experiment	-		HO-1, NQO1	CUR by the Nrf2/HO-1 axis could improve ethanol-induced liver oxidative damage.	[18]
Hepatic Fibrosis	CUR	Male ICR mice	100, 200, and 400 mg/kg, orally, once a day, 4 weeks	LX-2	10–40 µM	PPARa, C/EBPa	CUR via activating Nrf2 could induce lipocyte phenotype in HSCs.	[19]
Liver Injury	CUR	-	-	HSC-T6	0.15 µM	α-SMA, MDA, GSH,	CUR via upregulatingNrf2 could protect HSC-T6 cells against oxidative stress.	[20]
Nonalcoholic Fatty Liver Disease (NAFLD)	CUR	Male C57BL/6 mice	50 and 100 mg/kg, daily, orally, 4 weeks	Primary hepatocytes	10 μM	FXR, LXR, CYP3A, CYP7A, HNF4α	CUR via the Nrf2-FXR-LXR pathway could regulate endogenous and exogenous metabolism in NAFLD mice.	[21]
Nonalcoholic Steatohepatitis (NASH)	CUR	Males SD specific pathogens free rats	50 mg/kg/day, orally, 2 weeks	-	-	-	CUR via upregulating of the Nrf2 could play a role in treating NASH.	[22]
Mercury-Induced Hepatic Injuries	CUR	Wistar rat	100 mg/kg, daily, I P., 3 day	-	-	HO-1, ARE, γ -GCSh	CUR via the Nrf2-ARE pathway could play a role in protecting against mercury-induced hepatic injuries.	[23]
Arsenic-Induced Liver and Kidney Dysfunctions	CUR	Female Kunming mice	200 mg/kg, orally, twice a week, 6 weeks	-	-	MAPKs, NF-κB, HO-1, NQO1, JNK, ERK1/2, p38	CUR via inhibiting MAPKs/NF-κB and activating Nrf2 could function as an antioxidant and anti-inflammatory agent on arsenic-induced hepatic and kidney injury.	[24]
Paraquat-Induced Liver Injury	CUR, Nanocurcumin	Male Wistar rats	100 mg/kg, orally, daily, 7 days	-	-	HO1, NQO1	CUR via the Nrf2 could be supportive for the prevention and therapy of paraquat-induced liver damage.	[25]
Necrotising Enterocolitis (NEC)	CUR	Rat	20 and 50 mg/kg, orally	-	-	SIRT1, TLR4, NLRP3, Caspase-1	CUR via inhibiting TLR4 and activating SIRT1/Nrf2axis could improve NEC.	[26]

**Table 2 biomolecules-12-00082-t002:** Nrf2-related therapeutic effects of curcumin in ischemia-reperfusion (IR) injury.

Type of Disease	Type of Curcumin	Animals	Dose Range	Cell Line	Dose Range	Targets and Other Pathways	Function	Ref.
I/R-Induced Cardiac Damage	CUR analog 14p	Male C57BL/6 mice	Cur: 100 mg/kg/day, 14p: 10 mg/kg/day, given orally to the mice for consecutive 7 days before myocardial ischemia	H9c2	10 μM	Bax, Bcl-2, Caspase-3	CUR analog 14p via activating Nrf2 and decreasing oxidative stress could protect against myocardial I/R injury.	[27]
Hepatic I/R Injury	CUR	Male Albino rats	400 mg/kg, orally, daily, 14 days	-	-	HO-1, TNF-α, IL-1β, Il-6, iNOS	CUR via Nrf2/HO-1 activation could attenuate hepatic I/R injury.	[28]
Cerebral I/R Injury	CUR	Male Wistar rat	300 mg/kg, I.P., 30 min after occlusion	-	-	NF-κB	CUR via elevating Nrf2 and down-regulating NF-κB could reduce neurological dysfunction and brain edema after cerebral I/R.	[29]

**Table 3 biomolecules-12-00082-t003:** Nrf2-related therapeutic effects of curcumin in diabetes (I.P.: intraperitoneal).

Type of Disease	Type of Curcumin	Dose Range	Cell Line	Dose Range	Targets and Other Pathways	Function	Ref.
Diabetes-Related Cardiovascular Diseases	C66	5 mg/kg, orally, once a day in alternating days for 3 months	-	-	JNK2, TGF-β1, MCP-1, TNF-α, HO-1, SOD-1, Caspase-3	C66 via inhibiting JNK2 and upregulating Nrf2 could protect against diabetes-induced aortic damage.	[30]
Diabetes-Related Cardiovascular Diseases	CUR analog A13	20 mg/kg, daily, orally, 8 weeks	-	-	TGF-β1, NRF2, CAT, NQO1, COL1A2	CUR analog A13 via activating the Nrf2/ARE axis could ameliorate myocardial fibrosis in diabetic rats.	[31]
Diabetic Retinopathy (DR)	CUR	-	RPE	5–20 μM	HO-1, ERK1/2, Caspase-3	CUR via activating of the Nrf2/HO-1 axis could protect against HG-induced damage in RPE cells.	[32]
Diabetic Cardiomyopathy	CUR	100 mg/kg, daily, I.P., 6 weeks	-	-	HO-1, JAK, STAT, IL-6, NF-κB	CUR and metformin combination via Nrf2/HO-1 and JAK/STAT pathways could play a role in the treatment of diabetic cardiomyopathy.	[33]
Diabetic Nephropathy (DN)	CUR	-	NRK-52E	5–20 μM	HO-1, E-cadherin	CUR via activating of Nrf2 and HO-1 could protect renal tubular epithelial cells from high glucose (HG)-induced EMT.	[34]
Insulin-Resistant Conditions	CUR	50 mg/kg, daily, orally, 10 days	HepG2	10 μM	NQO-1	CUR via inhibiting inflammatory signaling-mediated Keap1 could upregulate the Nrf2 system in insulin-resistant conditions.	[35]

**Table 4 biomolecules-12-00082-t004:** Nrf2-related therapeutic effects of curcumin in nervous system disorders.

Type of Disease	Animals	Dose Range	Cell Line	Dose Range	Targets and Other Pathways	Function	Ref.
Traumatic Brain Injury (TBI)	Male ICR mice	50 and 100 mg/kg, I.P., 30 min after TBI	-	-	ARE, HO1, Bcl-2, Caspase-3, NQO1, Histone-3	CUR via the Nrf2-ARE axis could attenuate brain injury in the model of TBI.	[36]
TBI	Male C57BL/6 (wild-type, WT)	50 mg/kg Intraperitoneal injection, 15 min after TBI	-	-	Hmox-1, NQO1, GCLM, GCLC, Ccaspase-3, Bcl-2	CUR via the Nrf2 signaling could play neuroprotective roles against TBI.	[37]
Cerebral Injury	Kunming mice	200 mg/kg, orally, started 10 days before irradiation and continued for 31 days during radiation	-	-	NQO1, HO-1, γ GCS	CUR via enhancing the Nrf2 could ameliorate radiation-induced cerebral injury.	[38]
Neurotoxicity	-	-	Astrocyte	2–20 μM	ARE, HO-1, NQO1, Keap1	CUR via activating the Nrf2/ARE pathway independently of PKCδ could protect against MeHg-induced neurotoxicity.	[40]
Parkinson’s Disease (PD)	Male Lewis rat	100 mg/kg, twice a day for 50 days, orally	293T, SK-N-SH	-	HO-1, NQO1, AKT	CUR by activating the AKT/Nrf2 pathway could ameliorate dopaminergic neuronal oxidative damage.	[39]
Ethanol Associated Neurodegenerative Diseases	Male mice (C57BL/6N)	50 mg/kg, daily, orally, 6 weeks	HT22, BV2	2 µM	HO-1, TLR4, RAGE, GFAP, NF-κB, TNF-α, PARP-1, IL-1β, Bax, Bcl-2	CUR via Nrf2/TLR4/RAGE axis could protect the brain against ethanol-induced oxidative stress.	[41]
Diffuse Axonal Injury (DAI)	Male SD rat	20 mg/kg, I.P., 1 h after DAI induction	-	-	PERK, ATF4, CHOP, β-APP, eIF2α, CHOP, GSK-3β	CUR via the PERK/Nrf2 axis could mitigate neuronal cell apoptosis and axonal injury.	[42]
Quinolinic Acid-Induced Neurotoxicity	Male Wistar rats	400 mg/kg, daily, orally, 6 days	-	-	BDNF, ERK1/2, γ-GCL, G6PDH, GSH, SOD1, SOD2, CAT	CUR via BDNF/ERK1/2/Nrf2 could play a role in the treatment of quinolinic acid-induced neurotoxicity in Rats.	[43]
Chronic Unpredictable Mild Stress-(CUMS-) Induced Depression	Male SD rats	100 mg/kg, orally, daily, 4 weeks	-	-	ARE, NQO-1, HO-1, Nox2, 4-HNE, MDA, CREB, BDNF, PSD-95	CUR via activating the Nrf2 pathway could reduce CUMS-induced depressive-like behaviors.	[44]
White Matter Injury (WMI)	male Wistar rats	-	dorsal columns	50 μM	HO-1, NF-kB, ARE, HIF1-α, TNF-α, IL-1	Curcumin via Crosstalk between NF-kB and Nrf2 Pathways could exert a neuroprotective effect.	[45]

**Table 5 biomolecules-12-00082-t005:** Nrf2-related therapeutic effects of curcumin in renal disorders.

Type of Disease	Animals	Dose Range	Cell Line	Dose Range	Targets and Other Pathways	Function	Ref.
Kidney Injury	Male SD rats	100 mg/kg, orally, 5 consecutive days	293T	20 µM	HO-1, AKT Caspase-3, NF-ƙB, KIM-1	CUR and Thymoquinone combination via ameliorating Nrf2/HO-1 and attenuating NF-ƙB, KIM-1 could protect cisplatin-induced kidney injury.	[46]
Chronic Kidney Disease (CKD)	Human	320 mg/kg, daily, orally, 8 weeks	-	-	GPx, GR, SOD, GSH, GSSG MDA	CUR could reduce oxidative stress in nondiabetic or diabetic proteinuric CKD.	[48]
CKD	Male Wistar rat	120 mg/kg, orally, daily, 4 weeks	-	-	MCP-1, Nox-4, Dopamine D1R	CUR via the Keap1-Nrf2 axis could play a role in the treatment of CKD.	[47]
Passive HeymannNephritis (PHN)	Male Wistar and SD rats	300 mg/kg, orally, daily, 4 weeks	-	-	HO-1, PI3K, AKT, mTOR, p62, Bax, Caspase-3, Bcl-2, Beclin-1, LC3	CUR by regulating the Nrf2/HO-1 and PI3K/AKT/mTOR pathways could improve renal autophagy in experimental membranous nephropathy.	[49]

**Table 6 biomolecules-12-00082-t006:** Nrf2-related therapeutic effects of curcumin in cardio/pulmonary disorders (I.P.: intraperitoneal).

Type of Disease	Animals	Dose Range	Cell Line	Dose Range	Targets and Other Pathways	Function	Ref.
Idiopathic Pulmonary Fibrosis (IPF)	-	-	LMSCs	2.5–20 µM	HO-1, Bcl-2, Bax, Caspase-3, AKT	CUR via the AKT/Nrf2/HO-1 axis could protect murine LMSCs from H_2_O_2_.	[50]
High Altitude Pulmonary Edema (HAPE)	Male SD rat	50 mg/kg, orally, 1 h before exposure	A549	10 μM	HIF1-α, HO-1, VEGF, GST	CUR via upregulating Nrf2 and HIF1-α could play a role as a potential strategy for the prevention of HAPE.	[51]
Asthma	Female specific pathogen-free (SPF) BALB/c mice	200 mg/kg, I.P.,1 h prior to OVA	RAW264.7	5–50 μM	HO-1, NF-κB, TNF-α, IL-1β, IL-6	CUR via activating the Nrf2/HO-1 axis could ameliorate airway inflammation in asthma.	[52]
Chronic Heart Failure (HF)	Male C57BL/6 mice	50 mg/kg, daily, supplied with osmotic minipumps, for 8 weeks	-	-	HO-1, SOD2, myogenin, MyoD, MURF1, Atrogen-1,	CUR via upregulating the Nrf2 could ameliorate exercise intolerance in HF mice.	[53]

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
