# Peer review of "Nrf2-Related Therapeutic Effects of Curcumin in Different Disorders"

_biomolecules, 2022, doi:10.3390/biom12010082_

Round 1
Reviewer 1 Report
This review article summarizes the relationship between curcumin and Nrf2, along with its mediated signal molecules in different pathophysiological processes. However, this would be acceptable for publication after the major careful revision has been made as recommended below:
- Figure 1 illustrates the different physiological effects of Curcumin, but it seems to be rather monotonous, and thus should be optimized into more interests for a wide readership.
- There seems to be kind of certain repetition as described in lines 43-47. For example, "ischemic rehabilitation (IR) injury" and "cardiovascular diseases" belong to cardiovascular diseases.
- "Gastroenteric diseases" in line 54 are reviewed, but most of their contents are certain relevant to descriptions of liver injury (including Table 1). Thus, it is suggested to consider the consistent with the subtitle (such as digestive system diseases?).
- The dose of the drug used in the Tables needs to be unified. (μm? μM? μmol/L?)
- "THC" and "OHC" (as well as MACs in line 83, C66 (line 102) and A13 (line 110) emerged in line 55 and Table 1 should be defined by giving their chemical structures as Curcumin metabolites or analogues for readers' understanding.
- The purpose of "human or animal study" as classified in Table 1 is not clear, but rather does not seem necessary to be described, according to the content. In addition, there are too many words in the Tables, so the design and content should be simplified to become more concise and focused.
- Please specify what substance "DMF" in line 91 , 93 and 96 refers to.
- The abbreviation "P" in the Tables is explained in detail.
- All the other abbreviations presented elsewhere should also be defined clearly.
Author Response
- We edited Figure 1.
- We edited this part.
- We changed the subtitle accordingly.
- We have used mg/kg as the unified unit.
- We have added the formulation of mentioned analogues of curcumin.
- We omitted the mentioned column from table 1. We also modified tables design.
- We defined DMF.
- We defined the abbreviation "I.P" in tables.
- We defined all abbreviations.
Reviewer 2 Report
This is an important paper that attempts to propose curcumin as promising agent on Nrf2 pathway in different disorders.  
The study is interesting, but there are some points that require attention.
1.The study does not examine and discuss the curcumin bioavailability. Please specify
2. Which is the role of the microbiota in curcumin absorption? 3.The authors do not explain the reason why curcumin analogues are used, this point must be added.  
4.The discussion of the in vivo and in vitro studies should be placed in separate paragraphs and showed in different tables
  
Author Response
1. We added notes about curcumin bioavailability.
2. We added notes about the role of the microbiota in curcumin absorption.
3. We described curcumin analogues.
4. We discussed in vivo and in vitro studies in separate paragraphs. However, since some studies contained both types of assays and the number of tables in too high, we could not separate them in tables.
Round 2
Reviewer 1 Report
The present version of this review article should be acceptable for publication in this journal.
Reviewer 2 Report
The manuscript has been modified as requested, so it can be accepted in the present form